# Estimation of the Effects of the Cross-Head Speed and Temperature on the Mechanical Strength of Kenaf Bast Fibers Using Weibull and Monte-Carlo Statistics

**Richard Ntenga [1],*** , **Saïdjo SAÏDJO [1]** , **Tibi Beda [2]** and **Alexis Béakou [3]**

[1]  Laboratory of Simulation and Testing, The University Institute of Technology, The University of Ngaoundéré, Ngaoundéré P.O. Box 455, Cameroon; saidjbadou@gmail.com
[2]  Laboratory of Mechanics, Materials and Photonics, Faculty of Sciences, The University of Ngaoundéré, Ngaoundéré P.O. Box 455, Cameroon; bedti@yahoo.fr
[3]  Institut Pascal, UMR CNRS/UBP/SIGMA 6602, Campus des Cézeaux, BP 10448, F-63000 Clermont-Ferrand, France; alexis.beakou@sigma-clermont.fr
*  Correspondence: rntenga@gmail.com or richard.ntenga@univ-ndere.cm; Tel.: +237-678-66-0481

**Abstract:** Methods used by different researchers to evaluate plant fibers' (PFs) mechanical performance, show great variance in results. In this work, 320 single kenaf fibers of gage lengths 10 and 20 mm were tensile-tested using four speed levels (0.05; 0.5; 1 and 5 mm·min$^{-1}$). Sixty-three other specimens were also tested under three temperature levels (50, 100, and 150 °C). Mechanical characteristics, namely Young's modulus, tensile strength, and failure strain were determined. Estimation of the dispersion on the data was performed using Weibull and Monte-Carlo statistics. Results showed a low scatter for cross-head speeds of 0.05, 0.5, and 1 mm·min$^{-1}$, compared to 5 mm·min$^{-1}$ for the two gage lengths used. Monte-Carlo average failure strength values were found to be close to the experimental values. A drastic drop in the tensile strength was observed for the temperature of 150 °C for varying hold times. The reported findings are likely to be used in the elaboration of a tensile test standard on PFs.

**Keywords:** plant fiber; mechanical properties; Weibull methods; Monte-Carlo simulation; mechanical testing; heat treatment

## 1. Introduction

The use of plant fiber (PF) reinforced composites (PFRCs) in engineering applications has now become very attractive [1–3]. The automotive, building, and construction industries place a great deal of interest in these materials as they can be used for the design of structural members [4–7]. PFs are viewed as excellent alternatives to synthetic ones from an environmental point of view, for their neutral biodegradability and combustibility as well as for the non-release of harmful gases or solid residues [8]. Kenaf bast fibers, for instance, have been widely used as reinforcements for polymer matrices and plasters due to their greater lengths of 1.5–3 m [9–12].

PF properties are considerably influenced by their hierarchic composite microstructure and their suspected viscoelastic behavior. A good knowledge of their mechanical properties is therefore crucial in applications such as reinforcement of structural members. Nonetheless, owing to their composite and viscoelastic nature, the necessity arises to study the effect of cross-head speed and temperature on the mechanical performance of PFs.

However, in the literature, the test speeds used for PFs characterization differ from one researcher to another. Baley [13] used a speed of 1 mm·min$^{-1}$ to characterize the mechanical properties of flax fibers. Fan [14] applied a crosshead of 0.1 mm·min$^{-1}$ to tensile test elementary hemp fibers. Wang et al. [15] and Shahzad [16] used a cross-head speed of 0.5 mm·min$^{-1}$ to determine the mechanical properties of

bamboo and hemp fibers. Cissé [17] studied the hygro-mechanical behavior of hemp fiber and applied a speed of 0.005 mm.s$^{-1}$ to determine the mechanical properties of hemp fiber. Ntenga [18] determined the mechanical properties of the Rhecktophyllum camerunense (RC) fiber using a cross-head speed of 20 mm·min$^{-1}$. Negoudi et al. [19] applied a speed of 15 mm·min$^{-1}$ to the diss fiber. The wide dispersion of the results observed by the same author and by many others, during quasi-static tensile tests, is most often attributed to methods of characterization of mechanical properties.

In addition, the dispersion of the mechanical properties in quasi-static tensile materials as a function of the gage lengths has been the subject of several studies [15,20–23]. These authors reported on the decrease of mechanical properties with the gage length.

One of the most important factors is the cross-head speed. International testing standards [24] for yarns and ropes provide a range of speeds that can be used for all natural fibers. As mentioned above, PFs have a more complex microstructure and behavior [25] than synthetic fibers. Their mechanical properties are obviously influenced by the cross-head speed during quasi-static tensile tests. To the best of our knowledge, there exists no specific standard for PFs tensile tests. It is therefore important to carry out more in-depth studies on the influences of cross-head speed on the mechanical properties of PFs.

The effect of temperature on the mechanical performance of PFs in general and of kenaf fiber in particular was investigated [11,17,21,26]. Furthermore, the regions of cultivation, climatic conditions, and the age of the plants are, among others, factors that influence the mechanical properties of the plant fiber. However, there is a large variability of the diameter along a single fiber, which causes a scatter on the mechanical properties [27]. Thus, statistical approaches are essential to describe the wider scatter of mechanical properties, particularly to estimate the variation of the probabilistic resistance of kenaf fiber.

Recently, research studies using the Weibull distribution weakest link focused on the statistical analysis of the tensile strength of PFs such as flax [20], wool [22], bamboo [15], and palm fiber [27]. Yet, some studies have reported limitations of the two-parameter Weibull distribution as it fails to accurately predict the large scattering effects on tensile strength [20,28]. In principle, Monte-Carlo methods (MCM) can be used to solve any problem having a probabilistic interpretation. Monte-Carlo methods are mainly used in three problems such as disordered materials, classes optimization, numerical integration, and generating draws from a probability distribution. A number of interesting research works using MCM can be found in the literature dealing with strength of composite materials [29–31] and temperature distribution in a synthetic functional material behaviors [32]. To the best of our knowledge, little or no attention has been given to combining the Weibull and MC statistics to characterize the strength of PFs. Hence, this paper aims to compare the results obtained using the modified Weibull model and the combined Weibull–Monte-Carlo method to estimate the strength of PFs. The reported findings are likely to be used in the elaboration of a tensile test standard on plant fibers.

## 2. Experimental Methods

### 2.1. Cultivation and Extraction of Kenaf

The kenaf plant, or *Hisbiscus cannabinus* L., was harvested from an experimental private estate at Karna Manga, District of Mbé, Adamawa Region (Cameroon), at an average temperature of 25 °C. The kenaf plant was harvested in September, four months after cultivation. The fibers were chemically extracted in a sodium hydroxide (NaOH) solution at 5% after 90 min of immersion at 95 °C. The fibers were dried and cut to lengths of about 15 cm. They were abundantly washed with water and finally rinsed in a solution of acetic acid (1%) for 10 min and in distilled water. The fibers were dried in an oven at 40 °C for 48 h, the fiber/water ratio being equal to 1:20.

## 2.2. Materials

Hand-separated kenaf fiber bundles of two gage lengths (10 mm and 20 mm) were used. The Kenaf fibers were eventually hand-mounted on a paper frame [16,33] as shown in Figure 1 to prepare and prevent damage during testing of specimens. TCM Taiwan Technology Brand epoxy resin was used to stick the fibers to the paper frame.

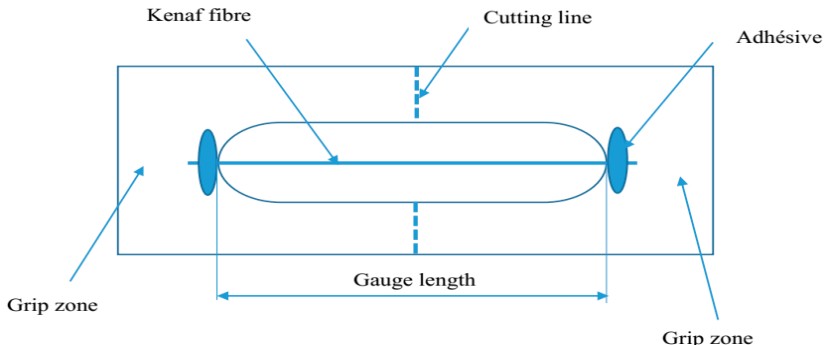

**Figure 1.** Tensile test and gripping tab specimens for kenaf fibers.

## 2.3. Tensile Strength of Kenaf Fibers and Heat Treatment

Quasi-static tensile tests were carried out on a universal machine, LIYI 1066A, equipped with a 500 N load cell, Type PST, accuracy class 0.05% F.S., controlled by proprietary software (TM2101) that records ultimate stress results. The machine was calibrated as needed and its compliance was determined before testing. Four cross-head speeds (0.05, 0.5, 1, and 5 mm·min$^{-1}$) were used to carry out tensile testing. Forty specimens were tested per length and per speed according to ASTM D3379-89 [24]. Outliers, namely specimens broken near the gripping tabs, were discarded. Twenty temperature and hold time dependent test specimens were prepared as described for speed dependent ones. Three temperatures levels (50, 100, and 150 °C) and three hold times (10, 30, and 60 min) were selected. Temperatures and hold times were chosen following PFRCs processing conditions [7,34]. The fibers were heated at prescribed temperatures and hold times, then cooled to room temperature before testing. Tensile tests were performed under standard environmental conditions (24 ± 1 °C and RH50 ± 1.5%). Scissor cuts were carefully made on both sides of the paper frame tabs at the mid-gage (Figure 1) just before the start of the test. Averaged diameter values of the fibers were computed from six-point measurements along the lengths as obtained with an Olympus CX43 optical microscope. Only Young's moduli of the fibers were estimated from a linear regression of the linear part at the end of the stress/strain curve.

## 3. Statistical Analysis

### 3.1. Two-Parameters Weibull Distribution Model

Statistical analysis methods are essential to study the wide dispersion of mechanical properties and to estimate the variation of the probabilistic failure strength of PFs. The two-parameter Weibull distribution, known as the WSGT [22,27], is the most widely used [15,22,23,27], and is given by:

$$P_f = 1 - \exp\left[-\frac{V}{V_0}\left(\frac{\sigma}{\sigma_0}\right)^m\right] \tag{1}$$

where $P_f$ is the probability of failure, $\sigma$ is the failure strength, $V$ is the volume of the fiber, $V_0$ is the standard volume, $m$ is the shape parameter (Weibull modulus), and $\sigma_0$ is the scale parameter.

The WSGT [22,27] can be rewritten as:

$$P_f = 1 - \exp\left[-\left(\frac{V}{V_0}\right)^\alpha\left(\frac{\sigma}{\sigma_0}\right)^m\right] 0 < \alpha < 1 \tag{2}$$

where $\alpha$ is the Watson–Smith parameter introduced [22] to represent variations of the diameter.

Once the Weibull distribution parameters ($m$, $\sigma_0$) are determined, the average value of the strength is obtained by the following relation:

$$\overline{\sigma} = \sigma_0\left(\frac{L}{L_0}\right)^{\frac{-\alpha}{m}}\Gamma\left(1 + \frac{1}{m}\right) \tag{3}$$

where $\Gamma$ is the gamma function, and $L_0$ is the reference length, generally normalized to 1 for mathematical convenience. In addition, the variance of $\sigma$ can be calculated as follows [35]:

$$\sigma^2 = \sigma_0^2\left(\frac{L}{L_0}\right)^{\frac{-2\alpha}{m}}\left\{\Gamma\left(1 + \frac{2}{m}\right) - \left[\Gamma\left(1 + \frac{1}{m}\right)\right]^2\right\}. \tag{4}$$

The development of Equation (2) can be found in references [15,21,22,27].

The parameter $\alpha$ is determined using the coefficient of variation $CV_D$ of the diameter, defined as [22]:

$$\ln(CV_D) = \alpha \ln(L) + A + \varepsilon \tag{5}$$

where $L$ represents the gage length, $A$ is a constant and $\varepsilon$ stands for the random error.

The probability of failure $P_f$ can be computed as average ranks as assigned to every measured strength value, $\sigma_{fi}$, for every gage length and diameter, and approximated as follow:

$$P_f = \frac{i}{N+1} \tag{6}$$

where $i$ is the ranks of strength points and $N$ the total number of batches.

### 3.2. Monte-Carlo Simulation

To account for the great dispersion encountered in the determination of the rupture stress, the Monte-Carlo simulation could be used. In practice this method is used to compare several mean values of the strength. Hence, for a given gage length, say $L_i$, and knowing the probability of failure $P_f$, the rupture stress is given by Equation (7) derived from the modified Weibull model (Equation (2)) as:

$$\sigma = \sigma_0\left\{\left(\frac{L_0}{L}\right)^\alpha \ln\left(\frac{1}{1 - P_f}\right)\right\}^{\frac{1}{m}}. \tag{7}$$

The random failure strength $\sigma$ can be obtained by generating a uniform random number $P_f$ in the interval [0,1]. A Matlab script was used herein to randomly produce 40 (as in the experiments) $P_f$ values in the interval [0,1], to estimate the ultimate stress. An average strength value was computed from each set of 40 random numbers. After $N$ number of simulations, the predicted strength was obtained by applying the criteria of the smallest standard deviation. $N = 100$ simulations were conducted accordingly. The tensile strength mean values used were estimated from modified the Weibull model (Equation (3)). As recommended by Virk et al. [36], multiple data sets (MDSs) based on multiple gage lengths and speed levels were used to account for accurate scaling predictions of the coupled Weibull–Monte-Carlo fiber strength estimate.

## 4. Results and Discussion

### 4.1. Influence of the Cross-Head Speed

In Figure 2 stress–strain curves are shown for each cross-head speed. As also reported in previous work by Baley [13] stress–strain curves at different cross-head speeds exhibit a non-linear zone at the very beginning, attributed to the reorientation of cellulose microfibrils about the axis of the fiber. The following linear zone represents the elastic behavior domain of the fiber. The Young's modulus needs to be estimated in this zone. Young's moduli of the fibers were estimated from a linear regression of the linear part at the end of the stress–strain curve. It could be observed that the stress underwent a linear increase with the strain until it reached its maximum value where rupture suddenly occurred.

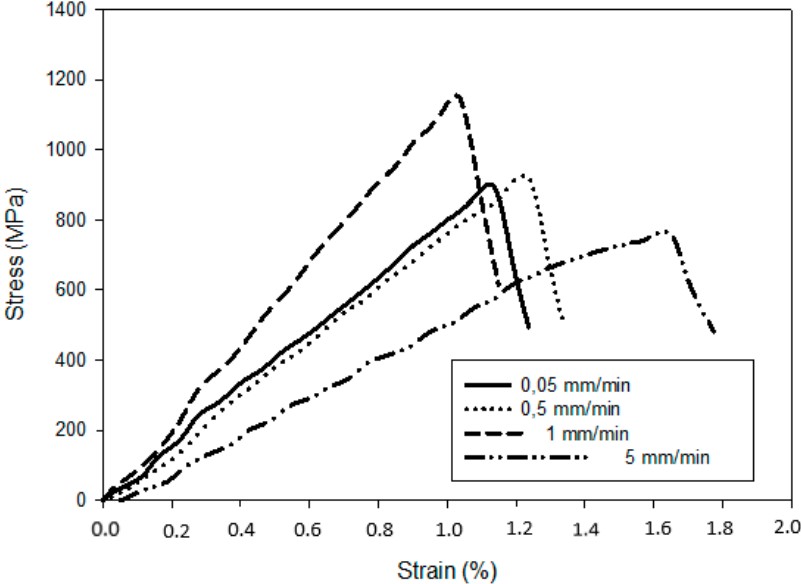

**Figure 2.** Tensile stress–strain curves for the four cross-head speeds at gage length 10 mm.

In addition, for the same gage length, a considerable effect of the cross-head speed on the fibers was observed, characterized by variable strains. Two trends could be reported from the test results: cross-head speeds 0.05, 0.5, and 1 mm·min$^{-1}$ exhibited a bilinear behavior at the beginning of the curve, marked by a two-slope pace, while the cross-head speed of 5 mm·min$^{-1}$ had a non-linear behavior, marked by a three-slope pace. Once again, the non-linear zone is often ascribed to the reorientation of cellulose microfibrils about the axis of the fiber. However, it could also originate from the viscous behavior of the fiber under tension. In fact, these assumptions were also confirmed by coupled X-ray diffraction and micro-tensile tests on hemp fibers by Placet et al. [37], modelling works by Charlet et al. [38] and Nilsson and Gustafsson [39] on flax and hemp fibers. Florent [40] suggested that the non-linear zone could be explained by existing non preconstrained fibers in the bundle. Thus, the stiffness of the observed bundle was characterized by a straightening of the fibers under tension. Test data were fitted to various probabilistic models from the XLStat software for the determination of mean values using the Kolmogorov Smirnov criteria that revealed the Log-normal distribution function as the best fitting model. Table 1 summarizes the derived values (stress, strain, and Young's modulus), per cross-head speed and gage length.

**Table 1.** Average values of mechanical properties of kenaf fibers at different cross-head speeds.

| GL (mm) | $\sigma_U$ (MPa) | | $\varepsilon_U$ (%) | | E (GPa) | | Speed (mm·min$^{-1}$) |
|---|---|---|---|---|---|---|---|
| 10 | 813.47 | ±41.57 | 1.46 | ±0.065 | 59.44 | ±2.94 | 0.05 |
| 20 | 617.18 | ±32.73 | 1.06 | ±0.054 | 63.32 | ±4.37 | |
| 10 | 800.53 | ±40.75 | 1.57 | ±0.079 | 54.81 | ±3.45 | 0.5 |
| 20 | 689.66 | ±34.78 | 1.29 | ±0.089 | 57.23 | ±2.54 | |
| 10 | 911.22 | ±43.29 | 1.62 | ±0.096 | 61.78 | ±3.33 | 1 |
| 20 | 710.83 | ±33.1 | 1.05 | ±0.05 | 70.87 | ±3.41 | |
| 10 | 808.16 | ±44.41 | 1.46 | ±0.063 | 59.33 | ±3.09 | 5 |
| 20 | 777.52 | ±47.44 | 1.17 | ±0.056 | 70.40 | ±4.42 | |

The results, as presented in Table 1, showed a large dispersion of the ultimate strength, the modulus of elasticity, and the ultimate strain for the four cross-head speeds and the two gage lengths. The highest value of the ultimate strength, 911.22 MPa, was obtained with the speed test of 1 mm·min$^{-1}$ and a gage length of 10 mm. However, as can be noted from the bar graph in Figure 3 (a plot of the data in Table 1), a significant effect of the test speed on the strength of PFs was manifested.

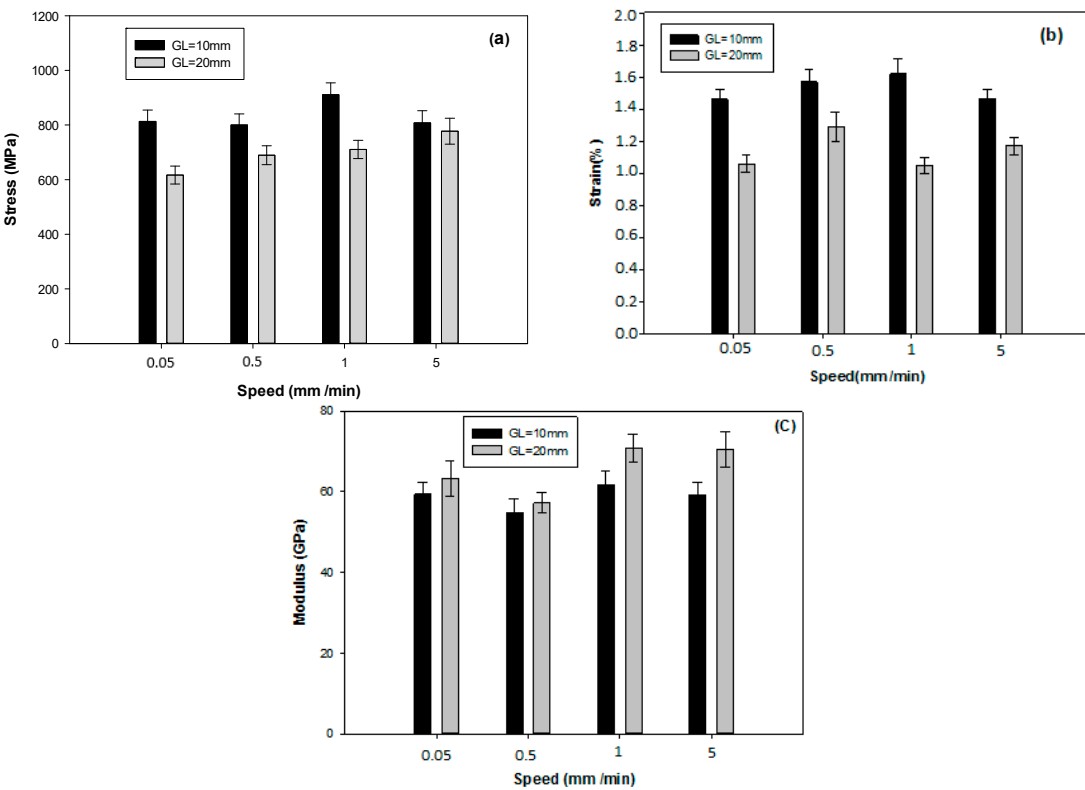

**Figure 3.** Tensile properties per test speed and gage length. (**a**) Stress data scatter, (**b**) Strain data scatter, and (**c**) Young's modulus data scatter.

The highest value of the tensile strength was reported for a test speed of 1 mm·min$^{-1}$ at 10 mm gage length. This value increased for all cross-head speeds at 20 mm gage length and was substantially constant for cross-head speeds of 0.05 mm·min$^{-1}$, 0.5 mm·min$^{-1}$, and 5 mm·min$^{-1}$ at 10 mm gage length. This trend was observed for both gage lengths. The values of strain for the four cross-head speeds were between 1.05 and 1.62%. These values were consistent with the literature [10,41,42]. However, it is worth noting that the lowest scatter was registered with the speed of 0.5 mm·min$^{-1}$ for the various properties and gage lengths.

### 4.2. Temperature Dependence

Table 2 summarizes the mechanical properties that were determined. Fibers with gage lengths of 10 mm, previously brought to different temperatures, were loaded at a cross-head speed of 1 mm·min$^{-1}$. The results showed a slight scatter on the Young's modulus and a wide scatter on the tensile strength for the three temperature levels and 'hold times' considered. For instance, for temperatures below 100 °C with 'hold times' of 10 to 30 min, the stiffness decreased from 61 to 32 GPa, whereas the stiffness decreased to 23 GPa at 150 °C for a 'hold time' of 60 min. The dehydration of the fibers was seemingly responsible for the brittleness that occurred, which caused a drop in mechanical properties observed for high temperature values. The fairly high values of the tensile strength of the fibers observed for the three temperature levels was ascribable to the presence of crystalline cellulose, a major chemical compound of kenaf material.

**Table 2.** Variation of mechanical properties as a function of temperature and temperature 'hold time (HT)'.

| HTs (min) | σ (MPa) | ε (%) | E (GPa) | T (°C) |
|---|---|---|---|---|
| | 793.2 ± 88.78 | 1.9 ± 0.26 | 36.02 ± 1.81 | 50 |
| 10 | 591.7 ± 83.71 | 1.8 ± 0.15 | 34.78 ± 3.36 | 100 |
| | 329.1 ± 41.03 | 0.9 ± 0.14 | 28.5 ± 2.92 | 150 |
| | 505.5 ± 62.54 | 1.6 ± 0.14 | 35.68 ± 2.62 | 50 |
| 30 | 495.7 ± 16.12 | 1.2 ± 0.07 | 31.70 ± 3.42 | 100 |
| | 234.8 ± 51.20 | 0.8 ± 0.11 | 26.16 ± 4.226 | 150 |
| | 477.2 ± 76.38 | 1.2 ± 0.39 | 25.27 ± 1.13 | 50 |
| 60 | 427.9 ± 60.39 | 1.01 ± 0.11 | 24.17 ± 3.69 | 100 |
| | 214 ± 23.26 | 0.6 ± 0.10 | 23.41 ± 2.59 | 150 |
| | 911.22 ± 43.29 | 1.62 ± 0.096 | 61.78 ± 3.33 | 24 |

Figure 4a–c, illustrate the scatter on the mechanical properties obtained from four temperature levels. A drastic drop was observed for the strength and Young's modulus for high temperatures (mainly for 150 °C), whereas for the strain, a slight decrease was reported for intermediate temperatures (50 and 100 °C). It appears that the exposure of fibers to high temperatures, followed by cooling at room temperature, induced an increase in the level of depolymerization and a decrease in the strength [43], as the fiber did not recover its initial mechanical properties. A drop in mechanical properties of flax fiber after 150 °C was reported by Gourier [23]. According to the author, these changes may be caused by thermal transition events within the chemical constituents (cellulose, hemicelluloses, lignin, and pectin) of the plant cell wall. A decrease in strength and the degree of polymerization of flax and jute fibers at temperatures above 170 °C was also reported by Gassan and Bledzki [43].

### 4.3. Two-Parameter Weibull Distribution Model

The data on mechanical properties, that is, average diameters and coefficient of variation per gage length and cross-head speeds, are shown in Table 3. The relationship between the coefficient of variation on diameter ($CV_D$) and the gage lengths allowed determination of the parameter $\alpha = 0.246$. As demonstrated by Zhang et al. [21], the value of $\alpha$ can be derived from the slope of the linear regression line of $ln(CV_D)$ versus $ln(L)$.

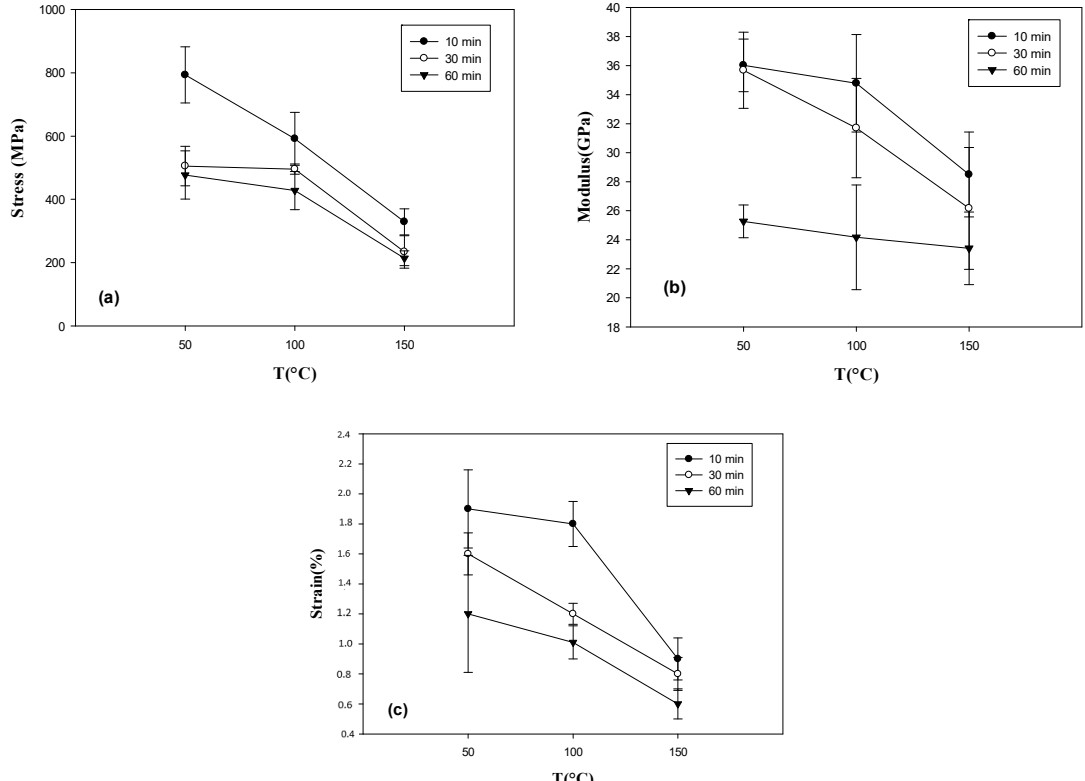

**Figure 4.** Tensile properties per temperature and temperature hold time. (**a**) Stress data scatter, (**b**) Young's modulus data scatter, and (**c**) Strain data scatter.

**Table 3.** Estimated Weibull parameters per gage length and cross-head speed.

| GL (mm) | Mean Diameter (μm) | $CV_D$ (%) | Weibull Module m | Scale Parameter $\sigma_0$ | Speed (mm·min$^{-1}$) |
|---|---|---|---|---|---|
| 10 | 60 ± 1.8 | 7.2 | 3.49 | 904.46 | 0.05 |
| 20 | 59 ± 2.1 | 8.6 | 3.40 | 689.04 | |
| 10 | 60 ± 1.8 | 7.2 | 3.47 | 891.46 | 0.5 |
| 20 | 59 ± 2.1 | 8.6 | 3.51 | 765.74 | |
| 10 | 60 ± 1.8 | 7.2 | 3.69 | 1010 | 1 |
| 20 | 59 ± 2.1 | 8.6 | 3.74 | 786.32 | |
| 10 | 60 ± 1.8 | 7.2 | 3.23 | 902.7 | 5 |
| 20 | 59 ± 2.1 | 8.6 | 3.23 | 870.21 | |

Figure 5 shows the Weibull fit of the tensile strength for the two gage lengths and four cross-head speeds. The Weibull modulus is determined as the slope of the regression line of the Weibull fit shown in Figure 5. As illustrated the linear regression of the failure strength at all gage lengths satisfactorily fit the experimental data with the values of $R^2$ ranging from 92.6% to 96.1%. The highest Weibull modulus, 3.74, was obtained at gage lengths of 20 mm and at the crosshead speed of 1 mm·min$^{-1}$, also at gage length of 10 mm and at the crosshead speed of 1 mm·min$^{-1}$. The smallest moduli were observed at gage lengths of 10 mm and 20 mm for the cross-head speed of 5 mm·min$^{-1}$. The estimated Weibull moduli of tensile strength, as seen in Table 3, were quite consistent with the experiments.

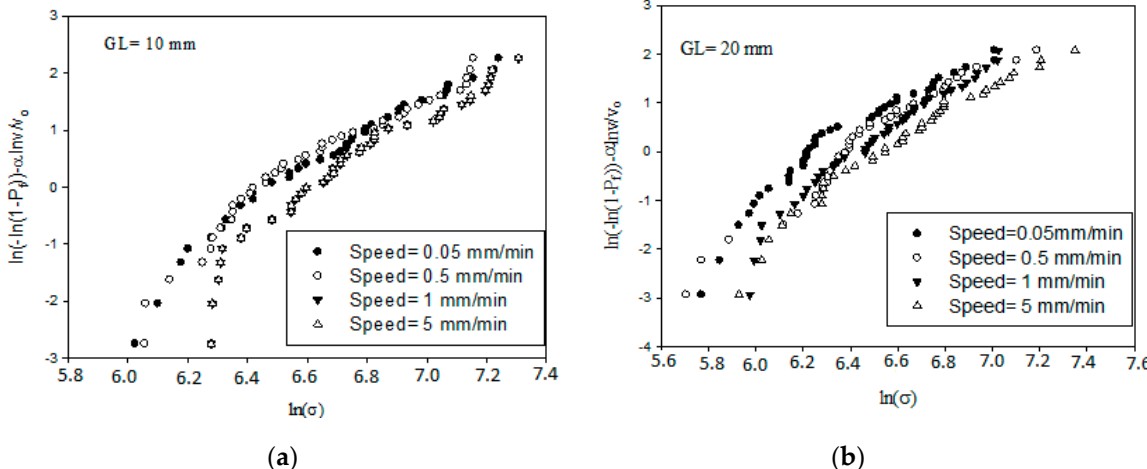

**Figure 5.** Weibull fit of the strength (**a**) at gage length 10 mm and (**b**) at gage length 20 mm.

### 4.4. Monte-Carlo Simulation Results of the Tensile Strength

Table 4 summarizes the average values of the tensile strength obtained from a Matlab script. From Equation (7) the parameters estimated by the modified Weibull model (Table 3) were used to simulate the average values of the tensile strength. Forty $P_f$ random numbers were generated in the interval [0, 1] and 100 simulations were implemented. The tensile strength mean values calculated were estimated from a modified Weibull model (Equation (3)). The objective of this section is to compare the mean tensile strength results obtained from both methods used herein.

**Table 4.** Average values of the simulated tensile strength from the estimated parameters of the Weibull model.

| GL (mm) | $\sigma_{exp}$ (MPa) | $\sigma_{cal}$ (MPa) | $\sigma_{rand}-\sigma_{simul}$ (MPa) | Speed (mm·min$^{-1}$) |
|---|---|---|---|---|
| 10 | 813.47 ± 41.57 | 686.16 ± 34.23 | 771.54 ± 29.49 | 0.05 |
| 20 | 617.18 ± 32.73 | 492.93 ± 25.13 | 546.22 ± 27.53 | |
| 10 | 800.53 ± 40.75 | 674.9 ± 33.86 | 759.07 ± 31.81 | 0.5 |
| 20 | 689.66 ± 34.78 | 553.03 ± 27.38 | 608.10 ± 23.67 | |
| 10 | 911.22 ± 43.29 | 776.16 ± 36.86 | 862.45 ± 32.02 | 1 |
| 20 | 710.83 ± 33.1 | 577.44 ± 27.02 | 643.80 ± 27.29 | |
| 10 | 808.16 ± 44.41 | 673.25 ± 35.96 | 807.93 ± 36.22 | 5 |
| 20 | 777.52 ± 47.44 | 613.84 ± 32.74 | 735.48 ± 35.48 | |

It was found that the modified Weibull model gave better results than the standard model [15,21,22] in evaluating the strength distribution of fibers at different gage lengths. However, as Figure 6 shows, experimental tests conducted on plant fibers exhibited the same trend as random Monte-Carlo experiments. As shown in Figure 6 (a plot of the data in Table 4), the failure strength predicted by the modified Weibull model combined with the Monte-Carlo simulation was more accurate than that obtained by the single modified Weibull model.

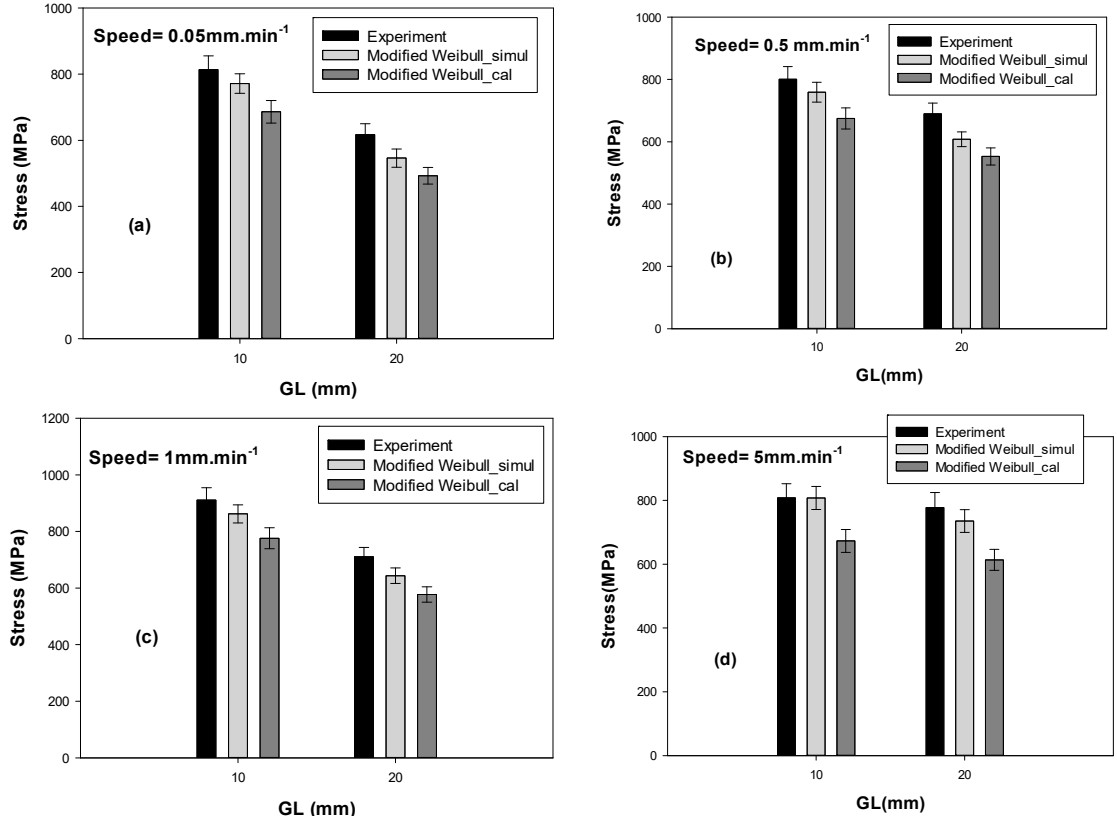

**Figure 6.** Experimental versus predicted average strength diagrams. (**a**) Tensile strengths at 0.05 mm·min$^{-1}$, (**b**) tensile strengths at 0.5 mm·min$^{-1}$, (**c**) tensile strengths at 1 mm·min$^{-1}$, and (**d**) tensile strengths at 5 mm·min$^{-1}$.

## 5. Conclusions

In this work, an evaluation of mechanical strength parameters and their temperature dependence of kenaf fiber at different test speeds and gage lengths was achieved using the modified Weibull model and modified Weibull model combined with the Monte-Carlo simulation. The statistics derived from the experimental data indicated that the cross-head speeds of 0.05 mm·min$^{-1}$, 0.5 mm·min$^{-1}$, and 1 mm·min$^{-1}$, depending on the two gage lengths (10 and 20 mm), caused less scatter of failure stress of the fiber compared with the speed of 5 mm·min$^{-1}$. The results obtained showed a decrease of the tensile strength of kenaf fibers with the increase of the gage length for the different test speeds. The use of the Monte-Carlo method combined with the Weibull statistical model permitted the estimation of the failure strength values very close to the experimental data. The effect of temperature and 'hold time' showed a significant decrease of the tensile strength at high temperatures and durations. The results obtained highlight the fact that PFs are very sensitive to temperature. Application of lessons learned herein to other PFs in future studies would definitely allow researchers to define a better practice of their tensile test.

**Author Contributions:** R.N. and S.S. designed the study and developed the theoretical framework. S.S. carried out the testing and wrote up the first draft of text as part of his Ph.D. dissertation. R.N. devised the project plan and supervised the research of S.S., T.B. and A.B. helped with interpreting the results and worked on the manuscript. All of the authors discussed the results and commented on the manuscript.

**Funding:** This research received no external funding.

**Acknowledgments:** One of the authors wishes to thank M.M. Ndam Njoya Arouna and Dangbe Ezekiel from the computer engineering department of the IUT, the University of Ngaoundere for their valuable and gracious help in some mathematical aspects of this work.

**Conflicts of Interest:** The authors declare no conflict of interest.

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
