# Peer review of "Estimation of the Effects of the Cross-Head Speed and Temperature on the Mechanical Strength of Kenaf Bast Fibers Using Weibull and Monte-Carlo Statistics"

_fibers, doi:10.3390/fib7100089_

Round 1

Reviewer 1 Report

The manuscript is well organized and contains good experimental results. The experimental techniques follow the international standards.

The manuscript deserves to be published in its present general form after moderate  English changes. 

Author Response

We have made few moderate revisions of the English language

Reviewer 2 Report

This paper presents a study on the mechanical properties of kenaf fibres at different gauge lengths, different cross-head speeds and different temperatures (at various hold times). The results are interesting, but unsurprising in nature, with cross-head speed affecting properties, as does gauge length. Temperature and hold-time are also found to be significant. The paper has merit and the results are important, but they could be presented significantly better; with error bars being absent from most of the graphs, but being essential for the correct interpretation of the data. I also noted a number of errors in the paper and have given a list of essential points below. One final thing is that the modified Weibull/Monte Carlo plots are not well explained to show how they differ from each other, and the claim that the results are very close to the experimental data seems to be dubious, so I would like further comment on that.

Essential changes:

Pg 1, line 36. “greater lengths” than what? What are you comparing the lengths to?

Pg 2, line 44 to 50. Consistent units for speed and also consistent formatting for the units. Note particularly line 45

Pg 2, line 52. Would 15 mm/min be considered as quasi-static loading? I would not test “quasi-statically” above about 1 mm/min, beyond that heating and inertia become significant and the speeds represent dynamic loading not quasi-static.

Pg 2, line 74. Should “inaccurately” actually be “accurately”?

Pg 5, Figure 2. How is this graph generated? Is it just four randomly selected samples from your data, or is it some form or average from all of the tests that you conducted? If it is the former, then please indicate how the data was selected. If it is the latter, then please include error bars on the data.

Pg 7, Figure 3. Please add error bars to your data so that the statistical significance of your data can be accurately assessed by the reader.

Pg 8, line 236. Is “plasticization” the correct word here? Normally plasticization would indicate an increase in the strain to failure, not a decrease. Would “embrittlement” be a better word?

Pg 11, Figure 6. Where are the error bars on this graph? Without error bars, the results do not look “very close”.

Pg 12. No mention of the effect of hold-time, which seems to be very significant in the data.

Pg 13, Ref 24. The title of this reference looks to be incomplete.

Reviewer 3 Report

Fig 3 needs to present experimental error values

Overall, this is a well designed and reported manuscript which provides interesting mechanical performance testing results.

Author Response

c)

We have made few and moderate revisions of the English language

Round 2

Reviewer 2 Report

The authors have responded to my criticism of the lack of error bars on the charts by saying that the standard deviation data is present in the tables from which the data is derived. I am very much aware of that, but still want to see error bars on the charts. One might as well say that we don't need the charts because the data is all in the tables. Error bars on charts help the reader to judge the statistical significance of the trends that are presented as bald numbers in the tables. From what I see in the tables, it appears that the errors overlap for some data that is claimed to be distinct, and some of the ones where it is claimed that the data is close shows no overlap of the errors. Please include error bars on the charts so that the reader can VISUALLY judge the data.

Author Response

Error bars are now added in figures 3 and 6. Many thanks for your vigilance.

Round 3

Reviewer 2 Report

Thank you for making the modifications.